# Optical electrophysiology for probing function and pharmacology of voltage-gated ion channels

Hongkang Zhang[1,2], Elaine Reichert[1], Adam E Cohen[1,2]*

[1]Departments of Chemistry and Chemical Biology and Physics, Harvard University, Cambridge, United States; [2]Howard Hughes Medical Institute, Harvard University, Cambridge, United States

**Abstract** Voltage-gated ion channels mediate electrical dynamics in excitable tissues and are an important class of drug targets. Channels can gate in sub-millisecond timescales, show complex manifolds of conformational states, and often show state-dependent pharmacology. Mechanistic studies of ion channels typically involve sophisticated voltage-clamp protocols applied through manual or automated electrophysiology. Here, we develop all-optical electrophysiology techniques to study activity-dependent modulation of ion channels, in a format compatible with high-throughput screening. Using optical electrophysiology, we recapitulate many voltage-clamp protocols and apply to $Na_v1.7$, a channel implicated in pain. Optical measurements reveal that a sustained depolarization strongly potentiates the inhibitory effect of PF-04856264, a $Na_v1.7$-specific blocker. In a pilot screen, we stratify a library of 320 FDA-approved compounds by binding mechanism and kinetics, and find close concordance with patch clamp measurements. Optical electrophysiology provides a favorable tradeoff between throughput and information content for studies of $Na_v$ channels, and possibly other voltage-gated channels.

*For correspondence: cohen@chemistry.harvard.edu

## Introduction

To gain detailed mechanistic insight into ion channel function and pharmacology, one often studies single channels, heterologously expressed, under voltage-clamp protocols (*Yan and Aldrich, 2012*; *Lewis and Raman, 2013*; *Bosmans et al., 2008*). Carefully designed sequences of voltage steps prepare channels in select conformational states (*Cordero-Morales et al., 2006*; *Zhang et al., 2012*). Distinct sub-states often have widely divergent affinities and kinetics of interaction with drugs. Knowledge of this state-dependent behavior is critical in developing models of channel function and in predicting how drugs will function in vivo. State-dependent dynamical measurements are typically the domain of manual or automated patch clamp. Functionally equivalent optical assays would open the prospect of high throughput screens with sophisticated state-dependent selection criteria; and might enable measurements in cell types or environments (e.g. in a tissue or whole animal) that are challenging to access with conventional methods.

Tools for optical electrophysiology—simultaneous optical perturbation and optical readout of membrane potential—have been making inroads into neuroscience (*Emiliani et al., 2015*; *St-Pierre et al., 2014*), with a primary emphasis on spatially resolved measurements in vivo (*Packer et al., 2015*; *Rickgauer et al., 2014*) or in complex cell cultures (*Hochbaum et al., 2014*). Voltage-sensitive dyes (VSDs) have been applied in a wide range of physiological assays in vitro (*Parsons et al., 1991*) and in vivo, (*Cohen and Salzberg, 1978*) but existing red-shifted VSDs are still excited by the wavelengths used to stimulate optogenetic actuators, leading to optical crosstalk (*Park et al., 2014*; *Tsuda et al., 2013*), or are not commercially available (*Huang et al., 2015*). A

**eLife digest** Ion channels are specialized proteins that span the cell membrane. When activated, these channels allow ions to pass through them, which can produce electrical spikes that carry information in nerve cells and regulate the beating of the heart. Researchers interested in understanding how ion channels behave often use a technique called patch clamp electrophysiology to measure the electrical current across the cell membrane. The technique can be used to probe if a specific drug can block an ion channel, but it is not well suited to screening lots of potential drugs because it is slow and expensive.

A group of ion channels known as voltage-gated sodium channels play an important role in generating the electrical spikes in nerve cells. One subtype called $Na_V1.7$ is involved in sensing pain and drugs that block $Na_V1.7$ might be useable as painkillers, but only if they are specific to this channel. This is because there are many similar sodium channels that are important in other processes in the body.

Zhang et al. have now developed a new light-based technique to measure how ion channels behave. The technique uses light to activate the channel and a fluorescent protein to report on the membrane's voltage. Zhang et al. used the new technique to probe how sodium channels, in particular $Na_V1.7$, interact with drugs. Mammalian cells grown in the lab were engineered to produce $Na_V1.7$, a light-activated ion channel (called CheRiff), and a fluorescent reporter protein. A flash of blue light delivered to the cells activated CheRiff, which in turn activated $Na_V1.7$. At the same time, the fluorescence of the reporter protein was used as a read-out of $Na_V1.7$'s activity.

Zhang et al. showed that they could reproduce many conventional electrophysiology measurements using their new light-based approach. Optical measurements were then used to screen 320 drugs to see whether they could block $Na_V1.7$. The results of the screen corresponded closely with measurements made using conventional electrophysiology. These results demonstrate that the new optical technique is both fast and precise enough to be used in drug discovery. Further studies could now ask if this optical technique can also be used to study other ion channels, such as potassium channels and calcium channels.

combination of a blue-shifted channelrhodopsin (CheRiff) and a red-shifted voltage indicator protein (QuasAr2) recently achieved spectrally orthogonal optical stimulation and readout (*Hochbaum et al., 2014*).

Optical electrophysiology measurements are typically semi-quantitative, at best, while ion channel assays require accurate perturbations to voltage and measurements of current. The optical techniques face several challenges: expression levels of optogenetic actuators and reporters vary from cell to cell; channelrhodopsins function as a conductance, not a voltage clamp; and fluorescence can only be used to measure membrane voltage, not current (*Cohen and Salzberg, 1978*; *Cohen et al., 1974*). Thus it is not obvious whether one can apply optical electrophysiology as a functional surrogate for standard voltage-clamp protocols.

Here we address this challenge by developing optical assays of the state-dependent electrophysiology and pharmacology of voltage-gated sodium ($Na_V$) channels. We begin with electrophysiologically inert HEK cells. We then stably express four transgenic constructs: an inward rectifier potassium channel and a voltage-gated sodium channel imbue the HEK cells with the ability to produce regenerative electrical spikes (*Hsu et al., 1993*; *Kirkton and Bursac, 2011*; *Park et al., 2013*). A channelrhodopsin variant, CheRiff, triggers these spikes upon exposure to flashes of blue light. An Archaerhodopsin variant, QuasAr2, enables fluorescent readout of membrane voltage via red excitation and near-infrared fluorescence. The QuasAr2 reporter has a ~1 ms response time and a linear response between −100 to +100 mV, providing a direct correlation of fluorescence and voltage (*Hochbaum et al., 2014*).

Brief flashes of blue light trigger sodium channel-mediated action potentials, which manifest as flashes of near infrared fluorescence. Steady state illumination with blue light induces steady state changes in voltage, and thereby changes in the distribution of $Na_V$ channels among substates. We develop stimulus and data analysis protocols that are robust to sources of cellular variation, and we

compare our results to measurements by manual patch clamp. While whole-cell voltage clamp remains the gold standard for absolute accuracy, optical electrophysiology provides a favorable tradeoff between accuracy and throughput.

Na$_V$ channels mediate the rising phase of the action potential and play significant physiological functions in excitable tissues. There are nine subtypes of Na$_V$ channels in the human genome. Na$_V$ channel dysfunction has been implicated in many human diseases. For example, loss-of-function mutations in Na$_V$1.1 can cause Dravet syndrome and in Na$_V$1.5 can cause Brugada syndrome (*Catterall, 2012*). The Na$_V$1.7 sodium channel plays an important role in mediating pain sensation. Homozygous loss of function leads to congenital insensitivity to pain (*Cox et al., 2006*), gain of function mutations lead to spontaneous severe pain, called erythermalgia (*Yang et al., 2004*), and nucleotide polymorphisms modify sensitivity to pain in the general population (*Reimann et al., 2010*). While recent results have suggested that the connection of Na$_V$1.7 to pain may involve other signaling pathways as well (*Minett et al., 2015*), there remains strong interest in finding selective blockers of this channel. Recent structural work has mapped an isoform-specific binding site for Na$_V$1.7-specific blockers (*Ahuja et al., 2015*), opening the possibility to develop new blockers via structure-guided design. Here we apply the Optopatch spiking HEK cell platform to study Na$_V$1.7, and we demonstrate its applicability to Na$_V$1.5 also.

## Results

### Construction and characterization of Na$_V$1.7 Optopatch Spiking (Na$_V$1.7-OS) HEK cells

We engineered a monoclonal HEK293 cell line stably expressing human Na$_V$1.7 and the Optopatch constructs (see Materials and methods). Both QuasAr2-mOrange2 and CheRiff-eGFP showed good membrane trafficking (*Figure 1B*). We used manual whole-cell patch clamp measurements to characterize the performance of each component. Under whole-cell voltage clamp (V$_m$ = −60 mV) CheRiff was activated by 488 nm light with an EPD50 (effective power density for 50% activation) of 20 mW/cm$^2$ and a saturating steady-state photocurrent density of 13.0 ± 1.2 pA/pF (mean ± s.e.m., $n$ = 5 cells, *Figures 2A,B*). As with channelrhodopsin 2, CheRiff showed inward rectification (*Gradmann et al., 2011*) with a reversal potential of +4 mV, consistent with non-selective cation conductivity. Under voltage steps from a holding potential of -100 mV, Na$_V$1.7 mediated robust inward currents with fast activation and inactivation kinetics within 10 ms and a peak current density of −61.4 ± 13.6 pA/pF at −20 mV (mean ± SD, $n$ = 11 cells, *Figure 2C*).

We found that stable expression of K$_{ir}$2.1 interfered with cell growth, so we expressed this channel via transient transfection. We call the quadruply expressing cells Na$_V$1.7 Optopatch Spiking (Na$_V$1.7-OS) HEK cells (*Figure 1A*). In a bath solution containing 2 mM K$^+$, Na$_V$1.7-OS HEK cells had a resting potential of −97.2 ± 2.2 mV (mean ± s.e.m., $n$ = 7 cell clusters), sufficient to prime most of the Na$_V$1.7 channels for activation. Upon voltage steps from −100 mV, K$_{ir}$2.1 showed inward rectifying behavior (*Figure 2D*).

Using manual patch clamp, we quantified the effect of CheRiff activation on membrane voltage. Brief optical stimuli (20 ms, 50 mW/cm$^2$) reliably triggered single spikes (*Figure 2—figure supplement 1A*), with a firing threshold of -48.0 ± 1.2 mV, peak depolarization of +30.1 ± 3.7 mV, and spike width at half-maximum repolarization (APD$_{50}$) of 33.5 ± 3.3 ms (mean ± s.e.m., $n$ = 5 cells). Under steady-state blue illumination, cells asymptotically approached a steady state depolarization that increased monotonically with stimulus intensity (*Figure 2—figure supplement 1B,C*), reaching an asymptotic value of –25.8 ± 6.2 mV (mean ± SD, $n$ = 4 cells) under intense illumination.

We then performed simultaneous recordings of membrane voltage and QuasAr2 fluorescence under optical CheRiff stimulation (*Figure 2E*). Cells were illuminated with continuous red light (640 nm, 400 W/cm$^2$) to excite QuasAr2 fluorescence. Pulses of blue light (500 ms on, 1.5 s off, stepwise increasing intensity from 1.1 to 26.0 mW/cm$^2$) were applied to activate CheRiff. Stimuli of intensity 15 mW/cm$^2$ or greater induced action potentials. The fluorescence traces faithfully reproduced both the action potential waveforms and the subthreshold depolarizations.

Optopatch measurements report membrane voltage, while patch clamp measurements typically control voltage and measure current. We thus used manual patch clamp measurements to determine the relation between voltage spike height measured in current clamp, and peak Na$_V$1.7 current

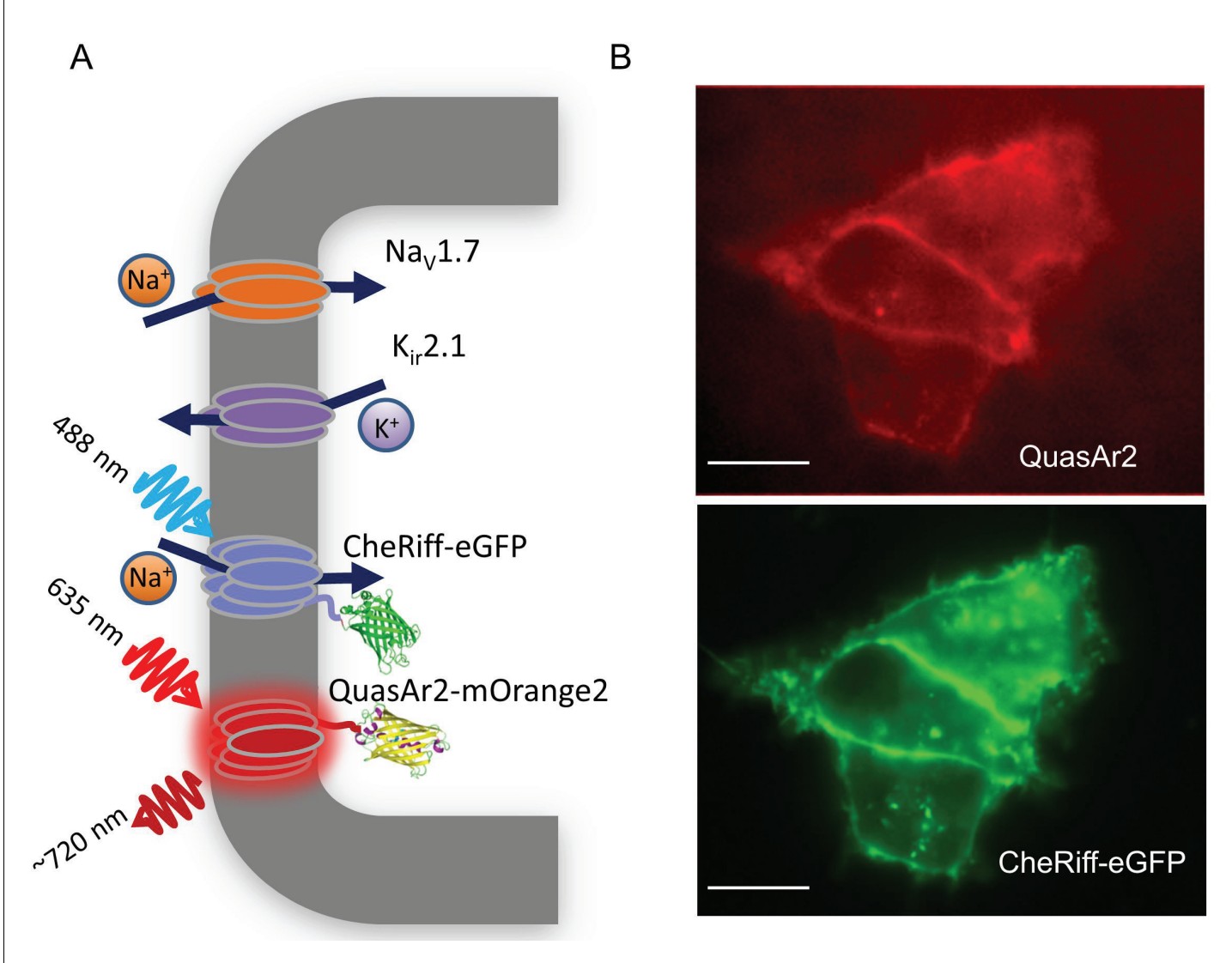

**Figure 1.** $Na_V1.7$ Optopatch Spiking ($Na_V1.7$-OS) HEK cells. (**A**) Genes expressed heterologously in $Na_V1.7$-OS HEK cells. $K_{ir}2.1$ maintains a hyperpolarized resting potential close to the $K^+$ reversal potential. $Na_V1.7$ imparts electrical excitability. CheRiff depolarizes the cells upon optical excitation and can trigger a $Na_V1.7$-mediated action potential. QuasAr2 is excited by red light and emits near infrared fluorescence in a voltage-dependent manner. (**B**) Epifluorescence images of QuasAr2 and CheRiff-eGFP expressed in $Na_V1.7$-OS HEK cells. Scale bar 10 μm.

density measured in voltage clamp. We used the state-dependent binding of amitriptyline to induce varying degrees of channel block, and then either applied a current pulse and measured the voltage response, or a voltage step and measured the current response (Materials and methods, *Figure 2—figure supplement 2A,B*). *Figure 2—figure supplement 2C* shows that the voltage spike amplitude was smoothly and monotonically related to the $Na_V1.7$ current density. Thus optical measurements of spike height are a quantitative probe of $Na_V$ current-carrying capacity. High-magnification fluorescence measurements showed that each individual cell gave a graded spike amplitude as a function of $Na_V1.7$ capacity (*Figure 2—figure supplement 3A*), with an 8% standard deviation in spike height at 50% channel block (*n* = 5 cells, *Figure 2—figure supplement 3B*).

Finally, we tested for photothermal or photochemical damage from the intense red illumination used for QuasAr2 imaging. In a 35 mm dish containing 2 mL of imaging buffer, continuous red illumination at 400 W/cm² (315 mW total power) for 10 min induced a temperature rise <0.3°C (*Figure 2—figure supplement 4A*). In a single well of a 384-well plate, containing 36 μL of imaging

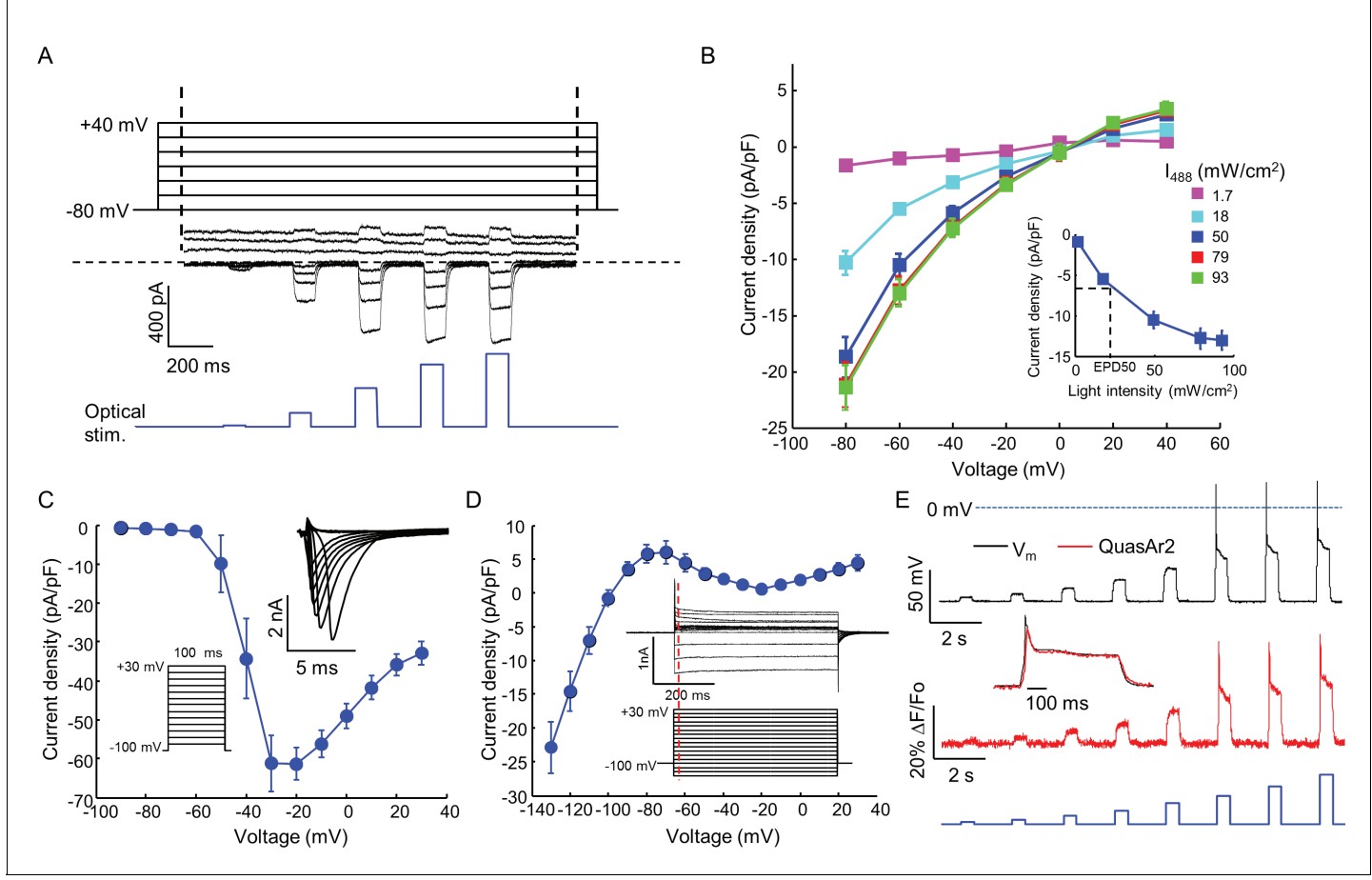

**Figure 2.** Biophysical characterization of $Na_V1.7$-OS HEK cells. (A) CheRiff current in a $Na_V1.7$-Optopatch HEK cell. Membrane potential was held at −80 mV and then stepped for 2 s to −80 to +40 mV in 20 mV increments. During each depolarization, the cell was exposed to 5 pulses of blue light, 100 ms duration, with increasing intensity (1.7, 18, 50, 79, 93 $mW/cm^2$). The horizontal dashed line indicates zero current. (B) I-V relation of CheRiff, under different light intensities. Currents were measured relative to baseline without blue light. Inset: Steady state photocurrent density as a function of blue light intensity, with a holding potential of −60 mV. (C) Peak $Na_V1.7$ current densities as a function of depolarization potential. Membrane potential was held at -100 mV and then stepped for 100 ms to −90 mV to + 30 mV in 10 mV increments. These measurements were performed prior to transient expression of $K_{ir}2.1$. Inset: currents in the 10 ms interval following each voltage step. (D) I-V relationship of $K_{ir}2.1$ expressed in $Na_V1.7$-OS HEK cells. Membrane potential was held at -100 mV and stepped for 500 ms to −130 mV to +30 mV in 10 mV increments. Inset: representative $K_{ir}2.1$ current recording. Red line indicates the time point (4 ms after voltage step) at which the current was quantified. (E) Simultaneous voltage and QuasAr2 fluorescence recording from $Na_V1.7$-OS HEK cells. The cell was exposed to a series of blue laser pulses, 500 ms duration, with increasing intensities (1.1, 2.3, 4.3, 7.0, 11, 15, 20, 26 $mW/cm^2$) and QuasAr2 fluorescence was monitored with 640 nm excitation, 400 $W/cm^2$. Inset: overlay of the voltage and fluorescence recordings from the most intense blue pulse (26 $mW/cm^2$).

The following figure supplements are available for figure 2:

**Figure supplement 1.** Current clamp recording of light triggered action potentials in Nav1.7-OS HEK cells.

**Figure supplement 2.** Relationship between $Na_v1.7$ current density and spike height.

**Figure supplement 3.** Cell-to-cell variability in Optopatch measurements.

**Figure supplement 4.** Effects of intense red laser illumination.

buffer, the temperature rise was 2.5 ± 0.4°C (mean ± s.e.m., *n* = 4 replicates) in 35 s and 9.8 ± 0.6°C (mean ± s.e.m., *n* = 4 replicates) in 10 min. While a 10°C rise is within the physiological range for measurements starting at room temperature (23°C), we kept all measurement protocols shorter than 35 s to avoid possibility of thermal artifacts.

Under these same conditions (400 W/cm$^2$, 384 well plate, 36 µL buffer) the cells continued to produce optically evoked spikes for 10 min, with little change in spike waveform (*Figure 2—figure supplement 4B*). The baseline QuasAr2 fluorescence dropped by 12% in this interval (*Figure 2—figure supplement 4B*) and the spike amplitude dropped from 3.5 ± 0.04% ΔF/F to 2.3 ± 0.06% ΔF/F (mean ± SD, n = 6 spikes, *Figure 2—figure supplement 4C*). The signal-to-noise ratio (SNR, spike height/baseline noise) dropped from 107 ± 2 to 73 ± 2.6 (mean ± SD, n = 6 spikes, *Figure 2—figure supplement 4C*). We explored the dependence of SNR on red illumination intensity (*Figure 2—figure supplement 4D,E*). At 400 W/cm$^2$ the SNR was 99.8 ± 3.3 (mean ± SD, n = 8 spikes) at a 100 Hz frame rate. At 6.3 W/cm$^2$ (5 mW) the SNR was 9.5 ± 1.1 (mean ± SD, n = 8 spikes) at the same frame rate. Thus spiking HEK cells can be imaged under a wide range of conditions, without photochemical or photothermal toxicity.

## Optically probing Na$_V$1.7 pharmacology with Na$_V$1.7-OS HEK cells

Sodium channel blockers are expected to change the firing properties of Na$_V$1.7-OS HEK cells. Most clinically used sodium channel blockers (e.g. lidocaine) show use-dependent or state-dependent action. We stimulated Na$_V$1.7-OS HEK cells with bursts of blue light (20 ms duration, six pulses) at 2, 4, and 8 Hz. The optically evoked action potentials were recorded by QuasAr2 fluorescence, averaging over ~150 cells. In untreated cells, each stimulus evoked an action potential. After addition of lidocaine (200 µM), cells continued to spike faithfully at 2 Hz. At 4 Hz and 8 Hz, cells spiked in response to the first stimulus, but failed for subsequent stimuli, a hallmark of activity-dependent Na$_V$ block (*Figure 3 A*).

Sodium channel blockers often show complex state-dependent binding affinities and kinetics. Voltage clamp protocols have been developed to prepare specific states to probe these mechanisms. Most voltage-clamp protocols comprise a prepulse, an optional recovery interval, and a test pulse. The voltage during each interval can be selected to populate different states. We sought to recreate these protocols by programming the duration and intensity of the blue light pulses.

First we varied the duration of the 488 nm optical prepulse from 20 ms to 500 ms, to probe state-dependent binding. The intensity was 50 mW/cm$^2$, which correspond to a depolarization to ~-30 mV. The cells were then given 200 ms recovery with no optical stimulus. The recovery interval was selected to allow drug-unbound channels to reprime. The test pulse (30 ms, 50 mW/cm$^2$) probed residual excitability. We used the amplitude of the fluorescence spike during the test pulse as a proxy for the degree of remaining Na$_V$1.7 current. Simultaneous fluorescence and manual patch clamp measurements showed close correspondence of the optical and electrical signals (*Figure 3B*). After adjusting the fluorescence data for scale and offset relative to the voltage recording (as in Ref. *Kralj et al., 2012*), the residual variations in fluorescence had an amplitude equivalent to 2.8 mV in a 200 Hz bandwidth.

We then applied the measurements using optical stimulation and recording alone, without patch clamp. Amitriptyline, a tricyclic antidepressant, showed strong state-dependent binding with degree of channel block dependent on prepulse duration (*Figure 3—figure supplement 1A*). The IC50 values of amitriptyline varied from 11.7 ± 1.6 µM at 20 ms prepulse to 1.6 ± 0.1 µM at 500 ms prepulse (standard error of fit to Hill equation, n = 3–5 wells per data-point, *Figure 3C*). This state-dependent block is consistent with previous patch clamp results (*Wang et al., 2004*). In contrast, TTX showed very modest state dependence (*Figure 3—figure supplement 1B*). The optically recorded IC50 values of TTX were 126 ± 13 nM at 20 ms prepulse and 62 ± 5.8 nM at 500 ms prepulse (standard error of fit to Hill equation, n = 4–6 wells per data-point, *Figure 3D*), consistent with prior findings that TTX has a slightly increased affinity for the inactivated channel (*Conti et al., 1996*; *Patton and Goldin, 1991*).Some sodium channel blockers can slow channel repriming after inactivation (*Bean et al., 1983*). We examined this effect by varying the recovery period. Cells were exposed to a prepulse with fixed duration of 500 ms, a variable recovery period from 40 ms to 5120 ms, and a test pulse of 30 ms. Cells treated with DMSO vehicle showed nearly complete recovery within 40 ms. Carbamazepine, a commonly used drug for the treatment of seizure and neurological pain, blocked recovery at 40 ms, but not at 80 ms. Amitriptyline had an even more dramatic effect, slowing the half-recovery time to 280 ± 36 ms, when tested at 3 µM (mean ± s.e.m., n = 9 wells per data-point). This result implies that amitriptyline has slow dissociation from the channel at resting membrane potential of -97 mV (*Figure 3E,F*).

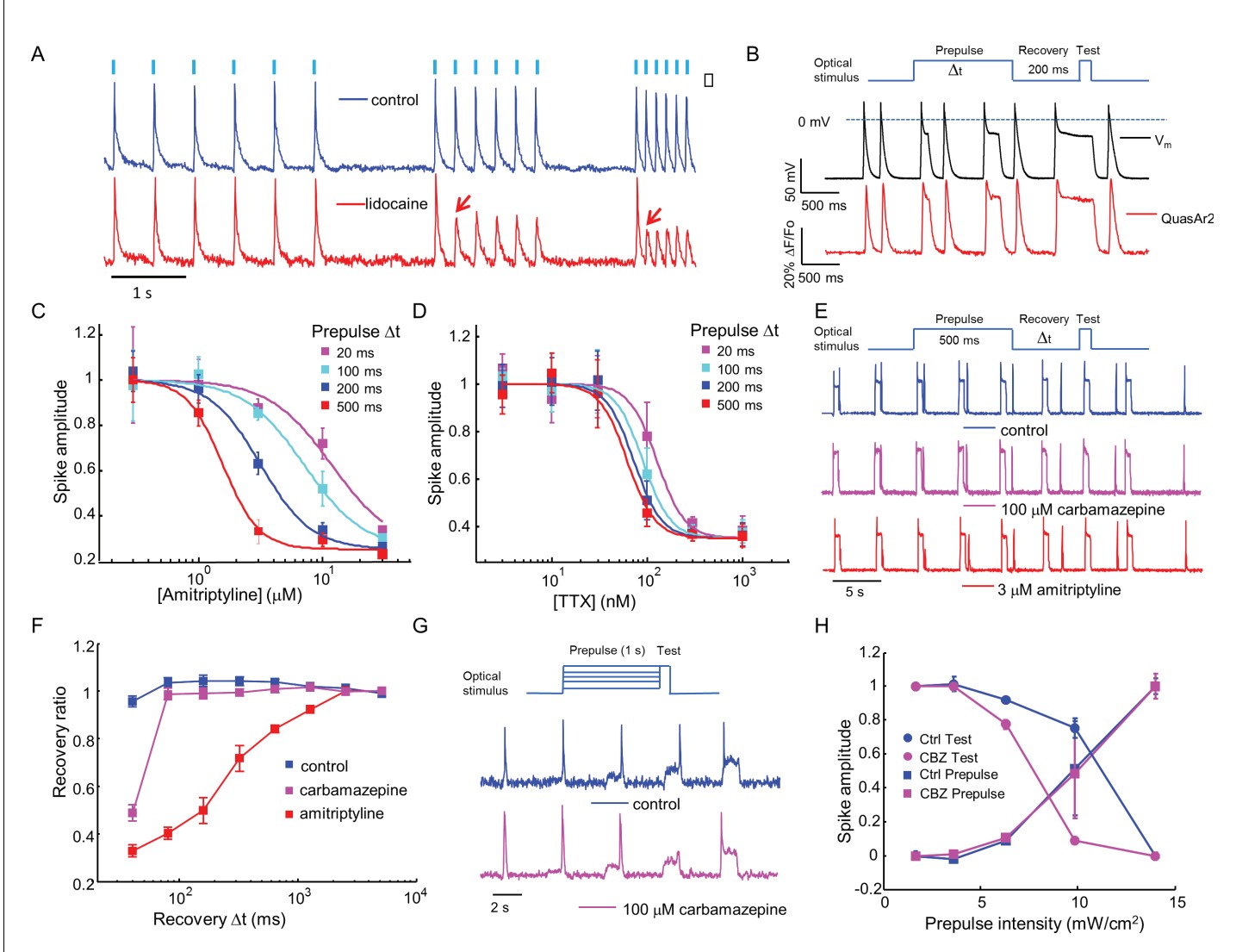

**Figure 3.** Mechanistic studies of Na$_V$1.7 blockers. (**A**) Approximately 150 Na$_V$1.7-OS HEK cells were stimulated with pulses of blue light (20 ms, 50 mW/cm$^2$) at increasing frequencies (2 Hz, 4 Hz, 8 Hz) and their total QuasAr2 fluorescence was recorded with 635 nm excitation, 400 W/cm$^2$. In control cells (blue), the fluorescence indicated spiking in response to each stimulus. After exposure to 200 μM lidocaine (red), cells showed activity-dependent block at 4 Hz and 8 Hz, but not at 2 Hz (red arrows). (**B**) Simultaneous current clamp and QuasAr2 fluorescence recordings from a Na$_V$1.7-OS HEK cell cluster (4 cells) stimulated with prepulses of varying length (20, 100, 200 and 500 ms; 50 mW/cm$^2$) followed by 200 ms recovery and a test pulse (30 ms, 50 mW/cm$^2$). (**C,D**) Application of the protocol in (**B**) to dose-response curves for (**C**) amitriptyline or (**D**) TTX. Test pulse spike amplitude was normalized to its value in the presence of the lowest tested concentration of drug (*n* = 3–5 wells for amitriptyline per data-point; *n* = 4–6 wells for TTX per data-point; ~150 cells per well). (**E**) Optical assay of Na$_V$1.7 recovery from fast inactivation. A 500 ms prepulse (50 mW/cm$^2$) populated the fast inactivated state and allowed drug binding. A variable recovery period (40 ms to 5120 ms) was followed by a 20 ms test pulse (50 mW/cm$^2$). Traces show fluorescence of QuasAr2 for control cells and after addition of either 100 μM carbamazepine or 3 μM amitriptyline. (**F**) Ratio of spike amplitude in test pulse to spike amplitude at the longest recovery time (5120 ms), as a function of recovery time. Carbamazepine modestly slowed recovery and amitriptyline dramatically slowed recovery (*n* = 9–11 wells per curve, ~150 cells per well). (**G**) Optical protocol to measure voltage-dependent Na$_V$1.7 activation and inactivation. Cells were stimulated with 1000 ms prepulse with increasing intensity (1.7, 3.6, 6.3, 9.8, 14 mW/cm$^2$), immediately followed by a test pulse (150 ms, 14 mW/cm$^2$). Traces show representative fluorescence recordings of control and 100 μM carbamazepine. (**H**) Effect of carbamazepine on activation and inactivation curves. Spike amplitudes were normalized to the maximum spike amplitude in the trace and were then plotted against prepulse intensity (*n* = 3 wells per curve). Carbamazepine left-shifted the inactivation curve, decreasing the optically measured overlap between activation and inactivation.

The following figure supplement is available for figure 3:

**Figure supplement 1.** Use-dependent inhibition of Na$_V$1.7 by amitriptyline and TTX.

Traditional voltage clamp protocols are flexible and precise in both time and voltage, while optical control of voltage is only semi-quantitative. Nonetheless, we explored stimulus intensity-dependent protocols, using the relation between steady state depolarization and illumination intensity (*Figure 2*, *Figure 2—figure supplement 1A*) as a guide. We developed a protocol to probe separately voltage dependent activation and fast inactivation of $Na_V1.7$. Cells were exposed to a pre-pulse of 1 s duration with variable intensity from 1.7 to 14 mW/cm$^2$, corresponding to depolarizations of 84 to $-47$ mV. The blue light intensity was then stepwise increased to 14 mW/cm$^2$, with no intervening recovery period. We quantified the amplitude of the fluorescence spike at the onset of the prepulse and the test pulse. The former probed channel activation, and the latter probed fast inactivation.

In conventional voltage clamp measurements, the region of overlap between activation and inactivation is called the 'window current' and is important in governing cellular excitability. Mutations that increase the window current have been associated with pain disorders (*Fertleman et al., 2006*) and cardiac arrhythmias (*Wehrens et al., 2003*). Compounds that decrease the window current by left-shifting inactivation or right-shifting activation have therapeutic potential (*Fertleman et al., 2006*; *Qiao et al., 2014*). We examined the effect of carbamazepine on voltage-dependent activation and fast inactivation. Consistent with observations from traditional electrophysiology (*Fertleman et al., 2006*; *Qiao et al., 2014*), carbamazepine reduced the overlap between activation and inactivation by leftward shifting the fast inactivation curve without altering the activation curve (*Figure 3G,H*). While optical electrophysiology is not able to quantify the window current, it can identify the qualitative mechanistic feature, the sign, and the approximate magnitude of the effect.

## Characterization of PF-04856264, a $Na_V1.7$ specific inhibitor

Recently, a subtype specific drug binding pocket was identified in the voltage sensor of Domain IV of $Na_V1.7$ (*McCormack et al., 2013*). Significant effort has gone into developing subtype-specific blockers, due to their potential analgesic applications. One such compound, PF-04856264, selectively blocks $Na_V1.7$ in a state-dependent manner, with reported IC50 of 28 nM when the steady state potential is $-70$ mV (*McCormack et al., 2013*). The mechanistic details of the interaction of this compound with the channel have not been characterized. We varied the precondition pulse length and found that even at 2 s of precondition pulse, PF-04856264 failed to inhibit the channel at 100 nM (*Figure 4A*).

We hypothesized that the discrepancy between our measurements and literature results was due to the difference between the resting potential of our cells ($\sim -97$ mV) and the steady state potential in the prior work ($-70$ mV). To control the resting potential of the $Na_V1.7$-OS HEK cells, we varied the extracellular $K^+$ concentration and observed approximately Nernstian behavior, as expected for a leak current dominated by $K_{ir}2.1$ (*Figure 4B*). We elevated bath potassium from 2 mM to 8 mM, which decreased the magnitude of the resting potential to $\sim$-70 mV. Under this condition and a 2 s prepulse, PF-04856264 inhibited $Na_V1.7$ mediated spikes with IC50 at 43 nM. We further investigated use dependent inhibition of $Na_V1.7$ by PF-04856264 with 8 mM bath potassium, and observed both tonic and use dependent inhibition (*Figure 4C*). Our results show that binding of PF-04856264 is strongly dependent on the resting voltage even at potentials where $Na_V1.7$ activation is minimal. Due to slow binding kinetics, sustained baseline depolarization is more effective than strong but brief depolarization at inducing binding.

## High throughput screening of $Na_V1.7$ inhibitors

$Na_V1.7$ is widely considered to be a promising target for analgesic drugs (*Dib-Hajj et al., 2013*), so we sought to develop a high throughput screen based on Optopatch measurements in $Na_V1.7$-OS HEK cells. We used the ability to optically stimulate and record to screen for activity-dependent modulators of $Na_V1.7$. The platform was based around a commercial inverted microscope (Olympus IX-71) with an automated scanning stage and an air objective. Optogenetic stimulation and fluorescence imaging were performed through the objective. Fluorescence from a region 320 by 166 µm, comprising approximately 150 cells, was binned on the detector and digitized at 100 Hz. We programmed the system to record sequentially from each well in a glass-bottomed 384 well plate.

We tested a library of 320 FDA approved drugs. Each well was treated with a single compound at 10 µM concentration. Amitriptyline (10 µM) and DMSO (0.1%) were used as positive and negative

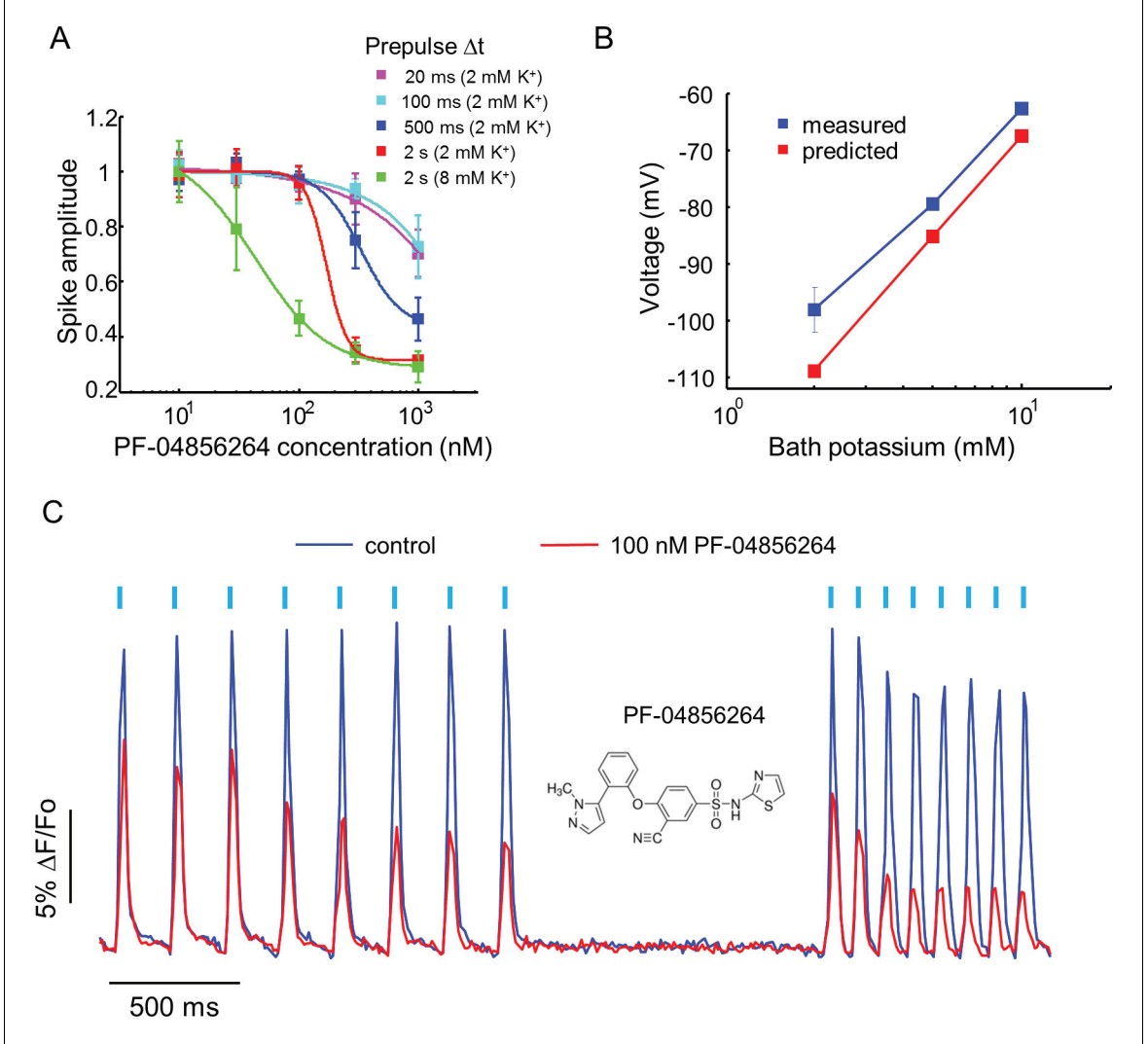

**Figure 4.** Effect of PF-04856264, a subtype-specific blocker, on Na$_V$1.7-OS HEK cells. (**A**) Dose-response curves for PF-04856264 when stimulated with prepulses of different durations and with different bath K$^+$ concentrations (*n* = 4 wells for each concentration). The optical protocol was as in *Figure 3B*, with prepulse duration specified in figure legends. (**B**) Comparison between membrane voltage predicted by the Nernst Equation (assuming pure K$^+$ conductance) and recorded by manual patch clamp, as a function of bath [K$^+$] (*n* = 4–7 cell clusters per data point). (**C**) Use-dependent inhibition of spiking in Na$_V$1.7-OS HEK cells by PF-04856264, at 8 mM external K$^+$. Cells were stimulated with eight pulses of blue light (20 ms, 50 mW/cm$^2$) at 5 Hz and 10 Hz and QuasAr2 fluorescence was monitored with 635 nm excitation, 400 W/cm$^2$. After photobleaching correction, the QuasAr2 fluorescence in the absence or in the presence of 100 nM PF-04856264, was normalized to peak amplitude of the first spike at 5 Hz in the absence of the drug. Each trace was averaged from 4 wells. Inset: structure of PF-04856264.

controls, respectively. Sixteen control wells (8 positive, 8 negative) were placed at the beginning and end of the plate. Cells were incubated with compound for 20 min. Each well was then stimulated with 8 pulses of blue light, 20 ms per pulse, 10 Hz, and the binned fluorescence was recorded. Automated scanning of the whole plate required 20 min.

Amitriptyline and DMSO showed robustly distinct firing patterns. DMSO wells showed consistent firing throughout the stimulus train. Amitriptyline wells showed rapid activity-dependent decrease in spike amplitude (*Figure 5A*). We also observed that some compounds induced more complex spiking patterns, either suppressing alternate spikes, or leading to erratic firing responses. We designed two simple parameters to capture the main features of use-dependent block, an important attribute of sodium channel blockers (*McCormack et al., 2013*; *Zhang et al., 2015*; *Scholz, 2002*). Let $S_i$ be the height of the i$^{th}$ spike (*i* runs from 1 to 8), and let $\tilde{S}_i \equiv S_i/S_1$ be the height of the i$^{th}$ spike divided

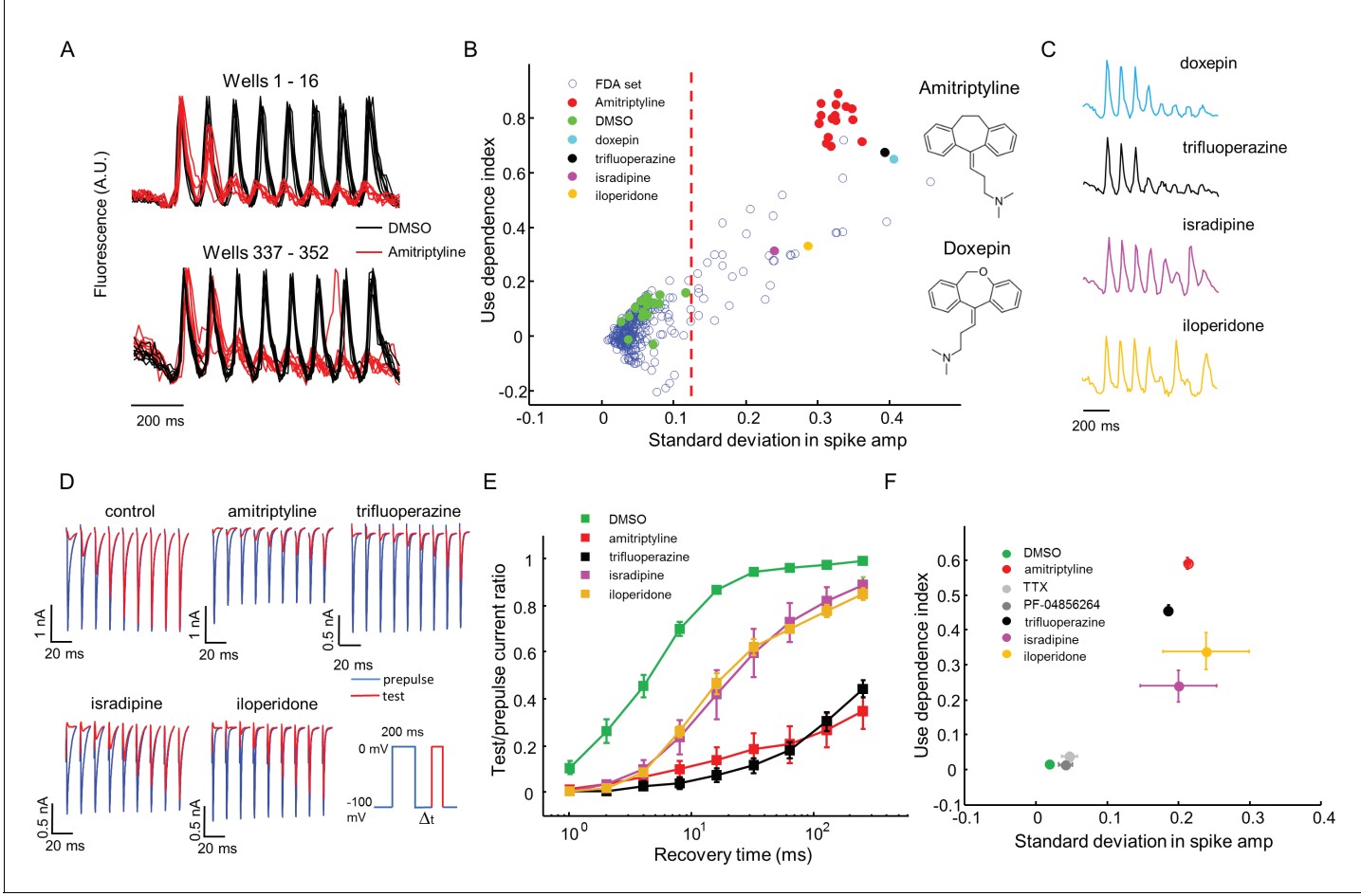

**Figure 5.** High throughput screening of a FDA-approved drug library in Na$_V$1.7-OS HEK cells. (**A**) QuasAr2 fluorescence from positive (amitriptyline) and negative (DMSO) control wells. Cells were stimulated with eight pulses of blue light (20 ms, 50 mW/cm$^2$) at 10 Hz, and QuasAr2 fluorescence was monitored with 635 nm excitation, 400 W/cm$^2$. (**B**) Screen results. The response of each well was parameterized by its use dependence index and standard deviation in spike amplitude. Positive controls (red) and negative controls (green) were well separated. Selected hits were chosen for further analysis. Inset: structures of amitriptyline and doxepin. (**C**) QuasAr2 fluorescence traces of doxepin, trifluoperazine, isradipine and iloperidone recorded in the screen. (**D**) Validation of select hits by manual electrophysiology. Na$_V$1.7-Optopatch cells were held at −100 mV. A 200 ms prepulse to 0 mV allowed drug binding. Recovery times at −100 mV ranged from 1 ms to 256 ms. A test pulse to 0 mV, 100 ms duration, probed the degree of channel recovery. Blue: Na$_V$1.7 current during prepulse. Red: Na$_V$1.7 current during test pulse. (Prepulse and test pulse currents have been time-shifted and overlaid for easy comparison). Each compound was tested at 10 µM. (**E**) Quantification of compound effects on Na$_V$1.7 recovery from inactivation. The plots show ratio of current amplitude at test pulse to prepulse, as a function of recovery period ($n$ = 3 cells for each compound, $n$ = 12 cells for control). (**F**) Characterization of select hits from (**B**) in Na$_V$1.5-OS HEK cells. Cells were stimulated with eight pulses of blue light (20 ms, 50 mW/cm$^2$) at 4 Hz. The 4 Hz stimulus was selected because action potential width of Na$_V$1.5-OS cells lasted longer than 200 ms under control conditions. Data analyzed and plotted as in (**B**) ($n$ = 4–6 wells per drug). Drug concentrations were TTX: 1 µM, PF-04856264: 1 µM, amitriptyline: 10 µM, trifluoperazine: 10 µM, isradipine: 10 µM, iloperidone: 30 µM).

The following source data and figure supplements are available for figure 5:

**Source data 1.** Spreadsheet containing compound names and screening results.

**Figure supplement 1.** Fluorescence traces from Na$_V$1.5-OS HEK cells with different drugs.

**Figure supplement 2.** Characterization of off-target effects via optical and manual patch assays.

by the height of the first spike (hence $\tilde{S}_1 = 1$). We calculated the use dependence index as $\Gamma = 1 - \left\langle \tilde{S}_i \right\rangle_{2-8}$, where the subscripts indicate the range of spikes averaged. We also calculated a measure of recovery from inactivation via the standard deviation in the spike amplitude, $\sigma = \left\langle \left( \tilde{S}_i - \left\langle \tilde{S}_i \right\rangle \right)^2 \right\rangle^{1/2}$. This parameter was large for wells that showed alternating or erratic spike patterns. Thus every well was represented by a point on a two-dimensional ($\sigma$, $\Gamma$) graph. Compound names and screening results are available in *Figure 5—source data 1*. Remarkably, in this blinded screen, Doxepin was classified as functionally adjacent to the amitriptyline controls. Doxepin is a tricyclic antidepressant with structure and pharmacology very similar to those of amitriptyline (*Figure 5B*).

Using $\sigma$ alone to distinguish positive and negative controls, the Z' factor (*Zhang et al., 1999*) for the assay was 0.57, within the range appropriate for a high-throughput screen (*Zhang et al., 1999*). We identified compounds for which $\sigma$ was greater than 5 standard deviations from the average of negative controls. The hit rate was 12.2% by this measure, consistent with the notion that voltage gated sodium channels are promiscuous binders (*Figure 5B*) (*Zhang et al., 2015*).

The 'hit' compounds showed diversity in their spike patterns. For amitriptyline, doxepin, and trifluoperazine, spike amplitude decayed monotonically throughout the pulse sequence. For other compounds, e.g. isradipine and iloperidone, spike amplitude alternated between even and odd stimuli, a pattern we called 'alternans' (*Figure 5C*). We hypothesized that the alternans compounds had a more transient inhibitory effect on channel repriming, compared to the amitriptyline-like compounds.

To test this hypothesis and to relate the parameters measured by the screen to more conventional electrophysiological parameters, we performed voltage clamp recordings in the presence of several hits from the screen. A 200 ms prepulse and a 100 ms test pulse were separated by a recovery interval varying from 1 ms to 256 ms. We measured the ratio of the peak inward sodium currents at the test and pre-pulse. Control cells showed half-maximal recovery in 4.5 ms. Cells treated with isradipine or iloperidone (alternans compounds), showed half-maximal recovery in 20 ms. Cells treated with amitriptyline or trifluoperazine (strong blockers) showed less than 50% recovery in 256 ms (*Figure 5D,E*). Thus compounds that clustered nearby in the optically measured ($\sigma$, $\Gamma$) graph, also showed similar effects by conventional patch clamp measurements.

For a compound to be a safe analgesic, it should not interact significantly with other $Na_V$ channels, particularly the cardiac $Na_V1.5$ channel. We created a $Na_V1.5$-OS HEK cell line analogous to the $Na_V1.7$-OS HEK cell line described above, and re-tested some hits from the screen against $Na_V1.5$ (*Figure 5—figure supplement 1*). Unsurprisingly, on a ($\sigma$, $\Gamma$) plot, all hits arranged in a similar pattern for $Na_V1.5$ and $Na_V1.7$, indicating no subtype selectivity ($Na_V1.7$-selective compounds are exceedingly rare). In contrast, both PF-04856264 (1 µM) and TTX (1 µM) clustered with the negative controls when tested on $Na_V1.5$, consistent with the known fact that neither of these compounds blocks $Na_V1.5$ at the concentrations tested (*McCormack et al., 2013*; *Zhang et al., 2015*) (*Figure 5F*).

Kir2.1 is not considered as a promiscuous drug binder (*Bowes et al., 2012*) and we are not aware of any compounds that block the conductance of channelrhodopsin or interfere with the voltage-dependent fluorescence of QuasAr2. Nonetheless, it is a formal possibility that false-positive readings could arise from compounds that modulate these other components. We developed a simple optical test for off-target effects and applied it to a panel of seven compounds selected for diverse mechanisms of $Na_V1.7$ block. First, we added TTX (1 µM) to block $Na_V1.7$. We then applied steps of blue light (500 ms, 3.2 – 56 mW/cm$^2$) and monitored the QuasAr2 fluorescence, which reported optically induced depolarizations without regenerative spikes (*Figure 5—figure supplement 2A*). We then performed the same measurement in the presence of TTX + test compound. Any drug interaction with Kir2.1, CheRiff, or QuasAr2 would alter the fluorescence response. Six of the seven tested compounds had minimal effect (within 7% of TTX, $n$ = 9–10 wells per compound). Carbamazepine induced a slight decrease in fluorescence signal (20% smaller than TTX alone, *Figure 5—figure supplement 2B*).

We further verified the optical tests using manual patch clamp measurements. None of the eight tested compounds (seven drugs and TTX) affected CheRiff photocurrents (*Figure 5—figure*

supplement 2C,D). Seven of the compounds had no effect on QuasAr2 fluorescence or voltage sensitivity relative to buffer control. Carbamazepine reduced QuasAr2 voltage sensitivity by 17%, consistent with the all-optical assay (*Figure 5—figure supplement 2E,F*). This off-target effect of carbamazepine had no effect on our optical assays of activity-dependent block (*Figure 3F* and *Figure 3G*), because fluorescence spike heights were normalized to the height of the highest spike. A slight decrease in overall fluorescence signal is cancelled in this analysis.

Finally, we explored whether the OS-HEK cells could be used to screen for modulators of other channel classes. $K_V4.3$ is an A-type fast-activating voltage-gated potassium channel, active in the heart and central nervous system. Conventional ionic flux based optical approaches to screening for modulators of $K_V4.3$ are extremely challenging because the channel is inactivated at the resting potential of HEK cells. We stably expressed $K_V4.3$, $Na_V1.5$ and Optopatch constructs in HEK cells. Voltage-clamp experiments revealed robust expression of $K_V4.3$, with a maximum current density of 218 pA/pF at +40 mV (*Figure 6A*). $K_V4.3$ has very fast activation kinetics with a time constant $\tau_{act}$ = 0.69 ms at +40 mV. The inactivation of $K_V4.3$ can be best described as a double exponential decay (*Liang et al., 2009*), with $\tau_{fast}$ = 51 ms and $\tau_{slow}$ = 352 ms at +40 mV (*Figure 6A*). We then transiently expressed $K_{ir}2.1$ to prime $Na_V1.5$ and $K_V4.3$ and called these cells $Na_V1.5$-$K_V4.3$-OS cells.

When stimulated by pulses of blue light (100 ms, 50 mW/cm$^2$), the presence of $K_V4.3$ led to a dramatic change in the optically induced and optically recorded action potential waveform, featuring a transient fast repolarization almost reaching resting potential before a recovery toward plateau potential (*Figure 6B*). We then tested the effect of heteropoda toxin 2 (HpTx2), a potent and specific blocker of channels in the $K_V4$ family (*Zarayskiy et al., 2005*). HpTx2 increased the action potential amplitude, consistent with its inhibition on the fast inactivated $K_V4.3$ peak current; HpTx2 also increased the plateau potential amplitude when compared at the end of the 100 ms light pulse, which can be explained by its inhibitory effect on the slow inactivated $K_V4.3$ current (*Figure 6C*). HpTx2 showed dose-dependent blockade, with an IC50 of 252 nM, consistent with literature results (*Figure 6D*) (*Brahmajothi et al., 1999*). However, the high rate at which test compounds blocked the $Na_V$ channel precluded use of these cells in high-throughput screening applications. Screening would require use of a $Na_V$ channel or a $Na_V$ channel mutant which is resistant to most drugs.

## Discussion

Despite variable expression levels of optogenetic actuator and voltage indicator, we have shown that Optopatch assays can probe state-dependent pharmacology of $Na_V$ channel modulators, and can accurately report binding affinities and kinetics. Key to achieving this accuracy were (1) performing measurements averaged over large numbers of cells, and (2) developing stimulus and analysis protocols that were insensitive to modest variations in expression levels of the optogenetic components.

Optical flux-based assays have been widely used in ion channel screens (*Yu et al., 2016*). However, these assays typically only probe steady-state channel behavior. Flux-based assays are widely used, however, because they offer high throughput and high reproducibility. Recent advances in automated electrophysiology (*Dunlop et al., 2008*) enable control of membrane voltage in heterologous expression systems. Automated electrophysiology offers the advantage of direct control of voltage and measurement of current. However these techniques have lower throughput and higher cost than optical assays, only work on certain cell types, and can be challenging to optimize. Patch clamp measurements also involve a perturbation to the integrity of the cell membrane, which can lead to changes in cytoplasmic composition and artifacts from mechanosensitive channels (*Morton and Main, 2013*). The Optopatch assays developed here provide detailed and quantitative mechanistic information; are compatible with high-throughput screening; and are non-invasive.

What are the limitations on throughput of optical electrophysiology screens? Here we performed serial measurements, one well at a time. At a measurement time of ~3 s/well, a 384-well plate was scanned in ~20 min. There are no fundamental principles that prevent scaling to more densely packed wells (e.g. 1534 well plates), or to parallelizing the measurements. While optopatch measurements require high intensity red illumination, high SNR can be obtained at lower intensities than the 400 W/cm$^2$ we used here (*Figure 2—figure supplement 4*). Specialized instrumentation has been developed for sensitive fluorescence recording from multi-well plates (*Hempel et al., 2011*), and with such instrumentation one could achieve throughputs compatible with primary screening. Given

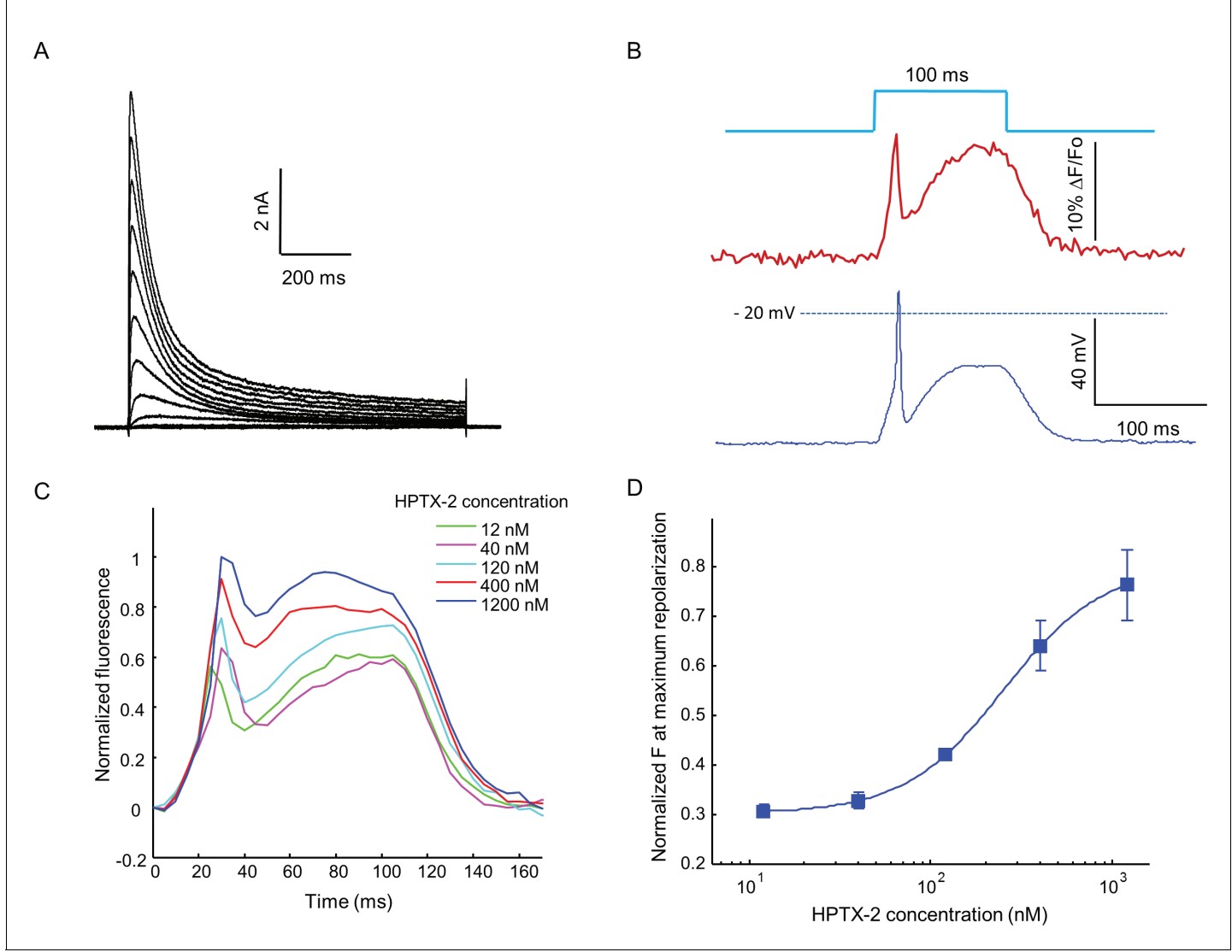

**Figure 6.** Optopatch assay of $K_V4.3$ function. (**A**) Voltage clamp recording of $K_V4.3$ current in $Na_V1.5$-$K_V4.3$ Optopatch HEK cells. The bath contained 30 μM TTX to block the $Na_V1.5$ current. Cells were held at −70 mV and then subjected to 1 s steps to −60 mV to +40 mV in 10 mV increments. Peak $K_V4.3$ current densities were 218 pA/pF. (**B**) $Na_V1.5$-$K_V4.3$-OS HEK cells were probed with simultaneous current clamp and QuasAr2 fluorescence. The cells were stimulated with a pulse of blue light (100 ms, 50 mW/cm$^2$), and QuasAr2 fluorescence was monitored with 640 nm excitation, 400 W/cm$^2$. $K_V$ activation led to a narrow action potential width, followed by $K_V$ inactivation and a return to steady-state depolarization. (**C**) Average QuasAr2 fluorescence traces from $Na_V1.5$-$K_V4.3$-OS HEK cells treated with HpTx-2 (*n* = 3–4 wells for each concentration). (**D**) Dose-response curve of HpTx-2 on $Na_V1.5$-$K_V4.3$-OS HEK cells. Drug effect was quantified by the fluorescence at the peak repolarization (~40 ms after onset of stimulus) relative to peak fluorescence intensity under 1200 nM HPTX2 treatment.

greater parallelism of measurement, one could also implement more complex stimulus protocols such as we developed here, while maintaining adequate throughput.

Illumination intensities for imaging Arch-based GEVIs are typically 10 to 100-fold greater than are used for imaging GFP-based GEVIs. Thus it is natural to worry about phototoxicity from the red laser. A recent study explored phototoxicity in cultured mammalian U2OS cells (*Wäldchen et al., 2015*). Illumination at 200 W/cm$^2$, λ = 488 nm for 240 s led to 100% of the cells being either dead or 'frozen'; while illumination at 5900 W/cm$^2$, λ = 640 nm for 240 s led to undetectable cell death. Our observation of good cell viability at 400 W/cm$^2$, λ = 640 nm is consistent with these literature results.

In principle, the screening approaches described here could be adapted to work with a red-shifted voltage-sensitive dye. Fluorescence signals would be more homogeneous than with a genetically expressed indicator; one could more readily switch between cell lines; and there is a possibility that the imaging could be performed at lower illumination intensity, on conventional equipment. However, existing red-shifted dyes, e.g. PGH1 (*Salama et al., 2005*) and Di-2-ANBDQPQ (*Zhou et al., 2007*) still retain considerable excitation at the blue wavelengths used for channelrhodopsin activation, and these dyes are not at present commercially available.

Finally, we consider the diversity of channels for which Optopatch-style screens may be feasible. Here we demonstrated assays for $Na_V1.7$, $Na_V1.5$, and $K_V4.3$. We previously demonstrated spiking HEK cells expressing $Na_V1.3$ (*Park et al., 2013*) and Hsu *et al.* demonstrated spiking CHO cells expressing $Na_V1.2$. HEK cell lines expressing $Na_V1.1$ through $Na_V1.8$ are commercially available, and a method for heterologous expression of chimeric $Na_V1.9$ was recently demonstrated (*Goral et al., 2015*). Voltage-gated $Ca^{2+}$ channels can also mediate regenerative spiking and thus are also plausible targets for the assay. Fast and repetitive optogenetic activation of $Ca_V3.2$ by channelrhodopsin2 has been achieved in HEK293T cells (*Prigge et al., 2010*). Recently, the state dependent inhibition of $Ca_V1.3$ has been studied by channelrhodopsin stimulation protocols (*Agus et al., 2015*). In principle, optogenetic activation could be applicable to other types of $Ca_V$ channels. Delayed rectifier potassium channels such as hERG and $K_V7$ may also be amenable to optical interrogation if co-expressed with an inactivation deficient $Na_V$ channel. Modulation of the potassium current would manifest as a change in the action potential duration (*Fujii et al., 2012*).

## Materials and methods

### Genetic engineering of $Na_V1.5$-OS, $Na_V1.5$-$K_V4.3$-OS and $Na_V1.7$-OS cells

The pIRESpuro3-$Na_V1.5$ and pcDNA3-$K_V4.3$ plasmids were obtained from ChemCORE at Johns Hopkins University. The Optopatch construct contains coding sequences of CheRiff-eGFP and QuasAr2-mOrange2, separated by a P2A self-cleaving peptide sequence. The entire Optopatch construct was cloned into a modified FCK lentivirus vector (mFCK), in which the original CaMKII promoter was replaced by a CMV promoter. The Kir2.1 cDNA was amplified from Addgene plasmid 32,669 (pENTR-L5-Kir2.1-mCherry-L2) and cloned into a pLX304 lentivirus vector that contained a blasticidin selection marker. The Kir2.1 cDNA was also cloned into pIREShyg vector using the Gibson assembly method. The Kv4.3 cDNA was amplified from pcDNA3-$K_V4.3$ plasmid and then cloned into pIREShyg vector using the Gibson assembly method (*Gibson et al., 2009*).

HEK293 cells were transected with pIRESpuro3-$Na_V1.5$ using TransIT-293 Transfection Reagent (Mirus Bio) following manufacturer's instruction. After 48 hr of transfection, puromycin was added to a final concentration of 2 µg/mL. Cells were selected for 14 days to stabilize the expression of $Na_V1.5$. Surviving cells were subsequently transduced with low-titer mFCK-Optopatch lentivirus. After 10 days of infection, all the GFP positive cells were enriched by fluorescence activated cell sorting (FACS). This polyclonal $Na_V1.5$-Optopatch stable cell line was used to generate the $Na_V1.5$-OS and $Na_V1.5$-$K_V4.3$ OS cells.

To generate $Na_V1.5$-OS cells, $Na_V1.5$-Optopatch cells were transduced by pLX304-Kir2.1 lentivirus. After 48 hr of transduction, $K_{ir}2.1$ expressing cells were selected by 5 µg/mL blasticidin. At the same time, 2 µg/mL puromycin was also included to ensure the stable expression of $Na_V1.5$. Cells were cultured for 14 days and then single cells were dispersed in wells of a 48 well plate. Monoclonal $Na_V1.5$ -OS lines were screened via Optopatch measurements for robust generation of action potentials under blue laser stimulus, and corresponding QuasAr2 fluorescence transients with SNR greater than 30.

To generate $Na_V1.5$-$K_V4.3$-OS cells, $Na_V1.5$-Optopatch cells were transiently transfected by pIREShyg-$K_V4.3$. Two days after transfection, 200 µg/mL hygromycin was used to establish the $Na_V1.5$-Optopatch-$K_V4.3$ monoclonal stable cell line. Each monoclonal cell line was optically evaluated for spiking and fast repolarization behavior after transient transfection of pIREShyg-$K_{ir}2.1$ plasmid. The best monoclonal cell line ($Na_V1.5$-Optopatch-$K_V4.3$) was further expanded and $Na_V1.5$-$K_V4.3$-OS cells can be reliably generated by transient transfection of Kir2.1 into this monoclonal cell line.

The $Na_V1.7$-OS HEK cells were generated based on a $Na_V1.7$ stable cell line established by G418 selection, a kind gift from Dr. Bruce Bean at Harvard University. This stable cell line was transduced with Optopatch by mFCK-Optopatch lentivirus. After 10 days, GFP positive ($Na_V1.7$-Optopatch) cells were enriched by FACS. We attempted, unsuccessfully, to further stabilize $K_{ir}2.1$ in these $Na_V1.7$-optopatch cells by using pLX304-$K_{ir}2.1$ lentivirus transduction. Surviving cells after blasticidin selection were not able to fire action potentials, likely due to poor expression level of $K_{ir}2.1$. Therefore, single cells of $Na_V1.7$-Optopatch cells were dispersed into a 48 well plate and each $Na_V1.7$-optopatch monoclonal cell line was evaluated by transient transfection of pIREShyg-Kir2.1 using lipofectamine 2000 (Invitrogen) following manufacturer's instruction. The best $Na_V1.7$-Optopatch monoclonal line that produced robust spikes with corresponding high SNR QuasAr2 fluorescence was selected and further expanded. The transfected cells are called $Na_V1.7$-OS cells.

Cells tested negative for mycoplasma contamination. Absence of contamination from other cell lines was ensured by growing up cells from a single clone.

## Cell culture

$Na_V1.5$-OS, $Na_V1.5$-Optopatch-$K_V4.3$ cells, and $Na_V1.7$-Optopatch HEK cell lines were maintained in Dulbecco's Modified Eagle Medium (DMEM) with 10% fetal bovine serum, penicillin (100 U/mL), streptomycin (100 µg/mL). For $Na_V1.5$-OS cells, 2 µg/mL puromycin and 5 µg/mL blasticidin were included in the medium to maintain expression of $Na_V1.5$ and $K_{ir}2.1$. For $Na_V1.5$-Optopatch-$K_V4.3$ cells, 2 µg/mL puromycin and 200 µg/mL hygromycin were included in the medium to maintain expression of $Na_V1.5$ and $K_V4.3$. For $Na_V1.7$-Optopatch cells, 500 µg/mL of G418 was included in the medium to maintain $Na_V1.7$ expression.

## Electrophysiology in HEK cells

Electrophysiology measurements were performed in a bath solution of Tyrode's, containing (in mM): 125 NaCl, 2 KCl, 2 $CaCl_2$, 1 $MgCl_2$, 10 HEPES, 30 glucose. The pH was adjusted to 7.3 with NaOH and the osmolality was adjusted to 305–310 mOsm with sucrose. Filamented glass micropipettes (WPI) were pulled to a resistance of 4–7 MΩ and filled with internal solution containing 140 mM KCl, 1 mM $MgCl_2$, 10 mM EGTA, 10 mM HEPES, 3 mM Mg-ATP, pH adjusted to 7.3 with KOH. To record CheRiff and $Na_V1.7$ current, $Na_V1.7$-Optopatch HEK cells were replated onto 0.02 mg/mL poly-d-lysine coated glass-bottom dishes (In Vitro Scientific) at a density of ∼10,000 cells/$cm^2$. The patch clamp recording was performed 4–8 hr after re-plating when most cells had firmly attached to the glass and were still dispersed as single cells. The whole cell voltage clamp recordings were acquired using an Axopatch 200B amplifier (Molecular Devices), filtered at 5 kHz with the internal Bessel filter and digitized with a National Instruments PCIE-6323 acquisition board at 10 kHz. The series resistance and membrane capacitance were compensated, and whole cell membrane capacitance was obtained by direct reading from the amplifier. CheRiff mediated current was triggered by Illumination from a blue laser (488 nm, 50 mW, Omicron PhoxX) that was sent through an acousto-optic modulator (AOM; Gooch and Housego 48058–2.5-.55-5W) for rapid control over its intensity. The Kir2.1 current was recorded from $Na_V1.7$-OS HEK cells by using the same configuration with 1 µM of TTX in the bath solution to block $Na_V1.7$ current. The $K_V4.3$ current was recorded from $Na_V1.5$-Optopatch-$K_V4.3$ cells with 30 µM of TTX in the bath solution to block $Na_V1.5$ current.

To correlate $Na_V1.7$ current density with voltage spike amplitude we performed alternate single-cell current clamp and voltage-clamp measurements in the presence of 3 µM amitriptyline. We used cells not expressing $K_{ir}2.1$ to avoid confound from $K_{ir}$ currents. Paired current- and voltage-clamp protocols were always performed on the same cell. For both protocols, an extended prepulse depolarization induced amitriptyline binding and complete channel block. A recovery interval at −100 mV of variable duration led to partial channel recovery. A test depolarizing pulse of either current or voltage then probed the response of the recovered channels.

In the current clamp protocol, holding current, $i_h$, was adjusted between −100 to −50 pA to attain a steady-state voltage of approximately −100 mV. The cell was then stimulated with a depolarizing current pulse of magnitude −0.5 $i_h$ for 500 ms. This current brought the steady-state voltage to ∼0 mV. The current was then brought back to $i_h$ for a recovery period of variable duration from 40–5120 ms. Finally, the cell was stimulated with a test current pulse of magnitude -0.5 $i_h$ for 20 ms to induce a voltage spike whose amplitude we recorded. Then the cell was switched to voltage-

clamp mode. The holding potential was −100 mV. To match precisely the degree of channel block in the current-clamp and voltage-clamp protocols, the voltage prepulse and recovery waveforms were copied directly from the voltage recorded during the immediately preceding current-clamp protocol. The test pulse comprised a 20 ms step depolarization to −20 mV. The inward $Na_V1.7$ current at each test pulse was then measured.

## Simultaneous electrophysiology and Optopatch recording in HEK cells

The $Na_V1.7$-Optopatch monoclonal cell line was transfected with pIREShyg-Kir2.1 plasmid using lipofectamine 2000 following standard protocols. The resulting Nav1.7-OS HEK cells were recorded 48 hr after transfection. The day before recording, cells were replated onto 35 mm glass-bottom dishes (In Vitro Scientific) at a density of ∼10,000 cells/cm$^2$. At the time when recording was performed, the cells formed small clusters comprising 3–4 cells. The whole cell current clamp recording was performed on these small clusters under the I-Clamp Normal configuration of the Axopatch 200B amplifier. The liquid junction potential was measured and corrected by the standard Neher method (*Neher, 1992*).

Patch clamp and fluorescence imaging data were synchronized by clocking the camera with analog output from National Instruments PCIE-6323 acquisition board while using the same clock for driving patch clamp inputs and outputs. The imaging experiments were conducted on a home-built inverted fluorescence microscope (*Hochbaum et al., 2014*). Briefly, QuasAr2 was excited by combined illumination from two red lasers (640 nm, 140 mW, Coherent Obis 637–140 LX and 640 nm, 100 mW, Coherent CUBE 640-100C) via a polarizing beam splitter. The red beam was expanded and focused onto the back focal plane of a 60× oil-immersion objective (60x APO, NA 1.49, Olympus). CheRiff was activated by Illumination from a blue laser (488 nm, 50 mW, Omicron PhoxX), which was modulated by an acousto-optic modulator receiving control signals from a National Instruments PCIE-6323 acquisition board. During a typical Optopatch experiment, both blue and red lasers were reflected into the sample plane by a quad-band dichroic mirror (Di01-R405/488/561/635-25x36, Semrock). The red laser intensity was maintained at 400 W/cm$^2$, while the blue laser intensity was modulated via the AOM and ranged from 1–100 mW/cm$^2$. A 710/100-nm bandpass filter (Chroma, HHQ710/100) was used for QuasAr2 imaging, and a variable-zoom camera lens (Sigma 18–200 mm f/3.5–6.3 II DC) was used to image the sample onto an EMCCD camera (Andor iXon Ultra 897), with 512× 512pixels. The variable zoom enabled imaging at a range of magnifications while maintaining the high light-collection efficiency of the oil-immersion objectives. Data were acquired with a ROI of 128 × 128 pixels at 4 × 4–pixel binning to achieve a frame rate of 200 frames/s.

## Optopatch measurements on pharmacology of Nav1.7-OS HEK cells

The $Na_V1.7$-Optopatch monoclonal cell line was transfected with pIREShyg-Kir2.1 plasmid using lipofectamine 2000. A glass-bottom 384-well plate (P384-1.5H-N, Cellvis) was treated with 0.02 mg/mL poly-d-lysine to promote cell adhesion. At 24 hr after transfection, cells were replated onto the multiwell plate at a density of ∼20,000 cells/well in 50 μL of culture medium. The imaging experiments were performed at 48 hr after transfection when the cells formed a confluent monolayer. The cells were washed with Tyrode's solution once and then each well is filled with 30 μL of Tyrode's solution. For drug additions, 6 μL drug solution at 6x target concentration was added to each well. Cells incubated in drug for 20 min at room temperature before imaging.

Experiments were conducted on an inverted epi-fluorescence microscope (Olympus IX-71) equipped with an automated scanning stage (Ludl electronics MAC 6000). Briefly, illumination from a red laser (635 nm, 500 mW, Dragon Lasers MRL-635-500 mW) was expanded and focused onto the back focal plane of a 20× air objective (NA 0.75, Olympus -UPlanSApo 20x/0.75). Illumination from a blue laser (473 nm, 50 mW, Dragon Lasers MBL-473-50mW) was sent through an acousto-optic tunable filter (AOTF; Gooch and Housego 48058) for rapid intensity modulation. The red illumination intensity at the sample was 400 W/cm$^2$. QuasAr2 fluorescence was filtered by a 710/100-nm bandpass filter (Chroma, HHQ710/100) and collected by an EMCCD camera (Andor iXon Ultra 897). Data were acquired with a full camera chip of 512 × 512 pixels at 16 × 16–pixel binning to achieve a frame rate of 100 frames/s.

## High throughput screening on Nav1.7-OS HEK cells by Optopatch measurements

Cells were plated in a 384 well plate as above. After the cells formed a confluent monolayer (48 hr after transfection) the cells were washed with Tyrode's solution once and then each well was filled with 20 µL of Tyrode's solution. A compound library consisting of 320 FDA-approved drugs was purchased from Broad Institute at 10 mM stock concentration in DMSO and then diluted to 30 µM in Tyrode's solution. 10 µL of the diluted compounds were added to the cell plate (Well A3-P22) to achieve a final concentration of 10 µM. Wells A2-H2 and A23-H23 were treated with 0.1% DMSO vehicle and used as negative controls. Wells I2-P2 and I23-P23 were treated with 10 µM amitriptyline and used as positive controls. After 20 min of drug incubation, the 384-well plate was placed on the microscope stage and each well was imaged serially.

The scanning started at well A2 and ended in well P23 in a column-wise manner. Each well was exposed to eight pulses (20 ms) of blue laser (50 mW/cm$^2$) at 10 Hz to stimulate the firing of Na$_V$1.7-OS HEK cells. The QuasAr2 fluorescence from each well was collected as above. Data were saved as a single tiff stack at the end of scanning.

## Imaging processing and data analysis

Imaging data were stored as a tiff stack and loaded into ImageJ software. For data acquired at high magnification (60×), a rectangular ROI surrounding the cells of interest was manually selected. Background fluorescence was determined by measuring the mean intensity of a nearby cell free region and was subtracted from the cell fluorescence. For data acquired from cell monolayers under low magnification (20×), a rectangular ROI (400 × 208 pixels) covering the region with most intense laser illumination was selected. This ROI corresponds to a 320 µm × 166 µm area on the sample plane, containing approximately 150 cells. The mean intensity within this ROI was calculated for all frames of the tiff stack. To calculate ΔF/F0, background fluorescence was determined by measuring the intensity from a well plated with parental HEK cells without QuasAr2 expression. After background subtraction, the data were further analyzed to extract spike parameters. Briefly, intensity traces were corrected for photobleaching by dividing the raw intensity by a median filtered copy of the intensity. Spike amplitude was defined as the difference between the maximum point of an action potential and the baseline. The use dependence index was defined as the fractional reduction of the spike amplitude averaged from the second to the eighth stimulus, compared to the initial stimulus.

Dose-response curves were fitted with the Hill equation y=START+(END-START)/[1+(IC50/S)$^n$], where START and END are the values of the parameter at minimum and maximum drug concentration, IC50 is the drug concentration at 50% maximum effect, S is the drug concentration, and n is a measure of cooperativity. The Z' factor for the screen was calculated as Z' = 1–3($\sigma_p$+$\sigma_n$)/|($\mu_p$-$\mu_n$)|, where $\sigma_p$ is the standard deviation of the positive controls, $\sigma_n$ is the standard deviation of the negative controls, $\mu_p$ is the mean of the positive controls and $\mu_n$ is the mean of the negative controls.

The activation time constant of K$_V$4.3, $\tau_{act}$, was determined by fitting the activation current trace using the equation: $i(t) = a \left(1 - e^{-t/\tau_{act}}\right)^4 + b$. The inactivation time constants, $\tau_{fast}$ and $\tau_{slow}$ of K$_V$4.3 were determined by fitting the inactivation current trace using the equation: $i(t) = a \ e^{-t/\tau_{fast}} + b \ e^{-t/\tau_{slow}} + \ c$.

## Statistics

Information on number of replicates for each experiment is given in figure legends. For manual patch clamp measurements, sample size was predetermined to be >5 cells, following standard practice. For optical electrophysiology measurements, sample size was predetermined to be >100 cells. These sample sizes were selected for feasibility of measurement. In the screen of the FDA library, one of the 32 control wells showed an anomalous spiking pattern (visible in *Figure 5A*) and was omitted from *Figure 5B*.

## Acknowledgements

We thank Bruce Bean for sharing the Na$_V$1.7 stable cell line, Melinda Lee and Katherine Williams for technical assistance, and Owen McManus for helpful discussions. This work was supported by the

Howard Hughes Medical Institute, and US National Institutes of Health (NIH) grant 1-R01-EB012498-01.

## Additional information

### Competing interests

AEC: A co-founder of Q-State Biosciences. The other authors declare that no competing interests exist.

### Funding

| Funder | Grant reference number | Author |
| --- | --- | --- |
| Howard Hughes Medical Institute | | Hongkang Zhang<br>Adam E Cohen |
| National Institutes of Health | 1-R01-EB012498 | Adam E Cohen |

The funders had no role in study design, data collection and interpretation, or the decision to submit the work for publication.

### Author contributions

HZ, Conception and design, Acquisition of data, Analysis and interpretation of data, Drafting or revising the article; ER, Acquisition of data, Analysis and interpretation of data; AEC, Conception and design, Analysis and interpretation of data, Drafting or revising the article

### Author ORCIDs

Adam E Cohen, http://orcid.org/0000-0002-8699-2404

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
