## [Decision Letter]

Thank you for submitting your article "Optical electrophysiology for probing ion channel function and pharmacology" for consideration by *eLife*. Your article has been reviewed by two peer reviewers, and the evaluation has been overseen by a Reviewing Editor and Richard Aldrich as the Senior Editor. One of the two reviewers has agreed to reveal his identity: Brian Salzberg.

The reviewers have discussed the reviews with one another and the Reviewing Editor has drafted this decision to help you prepare a revised submission.

Summary:

This manuscript describes the development of an all-optical electrophysiology technique for studying activity-dependent modulation of ion channels (in this case, Na_V_1.7), in a format compatible with high throughput screening. The authors employ expressed CheRiff as an optical activator, and QuasAr2 as an optical voltage reporter, in genetically engineered HEK cells. Their work demonstrates that optical electrophysiology provides, as they put it, "a favorable tradeoff between accuracy and throughput."

Essential revisions:

While the reviewers were favorable about the idea and many elements of the approach, they shared some concerns about the manuscript, which fall in to two categories: first, scientifically, no major new discoveries are clearly evident and second, technologically, some issues that are inadequately addressed. With respect to the first category, if the focus is indeed on the tool aspect, it might be possible to consider the manuscript for the Tools and Resources section of *eLife*, where the technique is more relevant than the discovery. But, in that case, the points in the second category become all the more important. The essential points to address therefore become the following:

The very high intensity light necessary to activate the optogenetic reporter and consequent questions of tissue damage and value over existing methods, expanded upon by the reviewers in point 1a and 1b below; 2) The apparent dependence of the technique on the regenerative capacities of V-gated Na channels, which might limit its general applicability, expanded upon by the reviewers in point 2 below, and 3) The issue of interference with other pharmacological agents, expanded upon by the reviewers in point 3 below.

1) The major criticism is over the use of QuasAr2 as the optical reporter. The light intensity used to excite QuasAr2 is about two orders of magnitude higher than (already high) intensities used normally for optogenetics. One wonders whether fluorescence excitation at an intensity of 400 W/cm2 is truly useful, or, for that matter, necessary. This illumination intensity is provided by not one, but two lasers, exciting the cells in parallel, because the quantum yield of the fluorescent protein is relatively low.

A) The bleaching and phototoxicity produced by such high illumination should be quantified. The authors should also provide some data on the heating effects of illuminating the preparation for extended times with 400 W/cm2 in the near infrared.

B) Also, why can't an organic voltage sensitive dye, applied in the bath, and requiring orders of magnitude lower illumination intensity be used as the optical reporter? There exist dozens of voltage sensitive dyes that can provide the speed (greater than that of QuasAr2), and the sensitivity to circumvent the illumination problem.

2) The authors claim their capacity to probe in a semi-quantitative way the activation, inactivation and state-dependent pharmacological blockade of voltage-gated sodium channels. However, the quantitative aspect of the measurements is questionable. The whole study is based on the occurrence of sodium-dependent regenerative depolarizations (spikes) produced by Na_V_ channels. The figures displaying single-cell recordings of these spikes (Figure 2 and Figure 2—figure supplement 1) show that they occur in an all-or-none fashion, as expected from a dynamical system with Na_V_ channels. Therefore, the appearance of partial blockade reported in the figures are obtained by averaging many cell, some of which are still able to generate spikes and others not. As a consequence, the range of modulation over which the technique is sensitive will depend heavily on the variability of Na_V_ expression. If all the cells were expressing the same density of Na_V_, one would expect response curves being step functions, as all the cells would cross the threshold for regenerative spiking at the same level of channel modulation. This aspect must be quantified extensively by monitoring the dependence of the spike on channel density and by measuring the distribution of Na_V_ channel expression in the polyclonal cell mixture, relative to the density threshold for regenerative spiking. Finally, the maximal plateau potential evoked by optogenetic stimulation looks dangerously close to spike threshold in Figure 2—figure supplement 1. This aspect must be quantified extensively. Additionally, while establishing a proof of principle for all-optical screening using this new optogenetic toolset, the paper falls short of comparing quantitatively all-optical measurements with electrophysiological measurements. Furthermore, the applicability to a wide range of voltage-gated channels, other than sodium channels, is not clearly demonstrated, given the dependence of the current assay on regenerative spiking.

3) The authors point rightly (Discussion, fourth paragraph) that interference of tested pharmacological compounds with the optogenetic reporters and actuators may produce false positive and false negative results, but that it is unlikely to happen. The burden of proof should not be left on potential users of their technique. The interference with the optogenetic tools of a wide subsets of the compounds used in this study, from different chemical families, should be tested using systematic patch-clamp recordings.

---

## [Author Response]

Essential revisions:

While the reviewers were favorable about the idea and many elements of the approach, they shared some concerns about the manuscript, which fall in to two categories: first, scientifically, no major new discoveries are clearly evident and second, technologically, some issues that are inadequately addressed. With respect to the first category, if the focus is indeed on the tool aspect, it might be possible to consider the manuscript for the Tools and Resources section of eLife, where the technique is more relevant than the discovery. But, in that case, the points in the second category become all the more important. The essential points to address therefore become the following:

We intended this submission for the Tools and Resources section, since indeed the emphasis is on the technique rather than a new scientific finding.

The very high intensity light necessary to activate the optogenetic reporter and consequent questions of tissue damage and value over existing methods, expanded upon by the reviewers in point 1a and 1b below; 2) The apparent dependence of the technique on the regenerative capacities of V-gated Na channels, which might limit its general applicability, expanded upon by the reviewers in point 2 below, and 3) The issue of interference with other pharmacological agents, expanded upon by the reviewers in point 3 below.

1) The major criticism is over the use of QuasAr2 as the optical reporter. The light intensity used to excite QuasAr2 is about two orders of magnitude higher than (already high) intensities used normally for optogenetics. One wonders whether fluorescence excitation at an intensity of 400 W/cm2 is truly useful, or, for that matter, necessary.

We added new text (subsection “Construction and characterization of Na_V_1.7 Optopatch Spiking (Na_V_1.7-OS) HEK cells”, sixth paragraph) and a new supplementary figure (Figure 2—figure supplement 4) characterizing in detail the effects of high-intensity red illumination on sample temperature, photobleaching, spiking behavior, and signal-to-noise ratio. We then studied the dependence of the signal-to-noise ratio on the red laser intensity. The new data and analysis establish that, while the red laser intensities seem high from the perspective of blue light excitation, the different spectral properties of cells in the red part of the spectrum place these experiments well within the biologically safe range.

We also added to the Discussion a description (third paragraph) of a recent study from the Markus Sauer lab which explored photochemical toxicity in cultured mammalian U2OS cells.^1^ Remarkably, illumination at 200 W/cm^2^, λ = 488 nm for 240 s led to 100% of the cells being either dead or “frozen”; while illumination at 5,900 W/cm^2^, λ = 640 nm for 240 s led to undetectable cell death. In our report we used illumination at 15-fold lower intensity and 6-fold shorter duration than the maximum 640 nm dose in the paper from Sauer et al. (Waldchen, S., Lehmann, J., Klein, T., van de Linde, S. & Sauer, M. Light-induced cell damage in live-cell super-resolution microscopy. Sci. Rep. 5, 15348 (2015); please see Figure 3 of this paper), which was proofed safe.

This illumination intensity is provided by not one, but two lasers, exciting the cells in parallel, because the quantum yield of the fluorescent protein is relatively low.

Some of our setups have two red lasers combined (a holdover from prior experiments), and some have only one. The screening experiments were performed on a setup with only a single red laser (described at the end of the subsection “Simultaneous electrophysiology and Optopatch recording in HEK cells”). The experiments can be performed with a single off-the-shelf diode laser that costs < $1,000.

A) The bleaching and phototoxicity produced by such high illumination should be quantified. The authors should also provide some data on the heating effects of illuminating the preparation for extended times with 400 W/cm2 in the near infrared.

We have included the requested data in Figure 2—figure supplement 4.

B) Also, why can't an organic voltage sensitive dye, applied in the bath, and requiring orders of magnitude lower illumination intensity be used as the optical reporter? There exist dozens of voltage sensitive dyes that can provide the speed (greater than that of QuasAr2), and the sensitivity to circumvent the illumination problem.

This critique does not offer a specific alternative voltage-sensitive dye (VSD), so we are left to guess what the reviewers have in mind. There are many interesting VSDs. We are not aware of any readily available bath-applicable VSDs that are sufficiently far red-shifted to avoid optical crosstalk with channelrhodopsin excitation. A recent effort at simultaneous optogenetic actuation and VSD imaging from the late David Yue illustrated the challenges of this approach:

Park, Sarah A., et al. "Optical mapping of optogenetically shaped cardiac action potentials." Scientific reports 4 (2014).

That paper used the red-shifted VSD PGH1. However, this dye retained sufficient excitation under blue illumination that voltage imaging was impossible during the epochs of optogenetic stimulation; and the SNR of the VSD measurements was much lower than what we report here.

Another recent effort is described in work from the George Augustine group:

Tsuda, Sachiko, et al. "Probing the function of neuronal populations: combining micromirror-based optogenetic photostimulation with voltage-sensitive dye imaging." Neuroscience research 75.1 (2013): 76-81.

This work used a red-shifted VSD, Di-2-ANBDQPQ, that also retains considerable absorption at the blue wavelengths used for channelrhodopsin excitation.

To our knowledge, neither PGH1 nor Di-2-ANBDQPQ is commercially available. We are not aware of any dyes that have the necessary spectral properties.

We added a paragraph to the Discussion (fourth paragraph) explaining that measurements with VSDs may offer advantages if the right dyes become available.

*2) The authors claim their capacity to probe in a semi-quantitative way the activation, inactivation and state-dependent pharmacological blockade of voltage-gated sodium channels. However, the quantitative aspect of the measurements is questionable. The whole study is based on the occurrence of sodium-dependent regenerative depolarizations (spikes) produced by Na*_V_
*channels. The figures displaying single-cell recordings of these spikes (Figure 2 and Figure 2—figure supplement 1) show that they occur in an all-or-none fashion, as expected from a dynamical system with Na*_V_
*channels. Therefore, the appearance of partial blockade reported in the figures are obtained by averaging many cell, some of which are still able to generate spikes and others not.*

The reviewers raise the important point that we did not quantitatively establish the relation between Na_V_ current, as would be measured by a conventional voltage-clamp assay, and voltage spike amplitude as measured in our assay. The fact that spike amplitude showed a step-like dependence on channelrhodopsin activation (Figure 2) does *not* necessarily imply a similar step-like dependence on Na_V_ conductance.

To address this critique we performed conventional patch clamp measurements in single HEK cells, alternately in current clamp and voltage clamp. We used the state-dependent Na_V_ block of amitriptyline (Figure 3) to induce varying degrees of Na_V_ block, and then measured either the spike height in current clamp, or peak Na_V_ current in voltage clamp. The new Figure 2—figure supplement 2 shows the relation of spike height to Na_V_ current (subsection “Construction and characterization of Na_V_1.7 Optopatch Spiking (Na_V_1.7-OS) HEK cells”, fifth paragraph). Even at the single-cell level, modulation of the Na_V_ current led to smooth and monotonic variation in the spike height. This result is essential for relating our results to conventional voltage-clamp experiments, and we thank the reviewers for spurring us to do this.

We performed additional Optopatch experiments at high magnification, repeating the amitriptyline measurement (Figure 3) with single-cell resolution. The new Figure 2—figure supplement 3 shows that optically recorded spike height in each individual cell is a continuous function of Na_V_ capacity, and that there is little cell-to-cell variability in this parameter.

*As a consequence, the range of modulation over which the technique is sensitive will depend heavily on the variability of Na*_V_
*expression. If all the cells were expressing the same density of Na*_V_*, one would expect response curves being step functions, as all the cells would cross the threshold for regenerative spiking at the same level of channel modulation. This aspect must be quantified extensively by monitoring the dependence of the spike on channel density and by measuring the distribution of Na*_V_
*channel expression in the polyclonal cell mixture, relative to the density threshold for regenerative spiking.*

The cells are *monoclonal* in Na_V_1.7, CheRiff, and QuasAr. To quantify the cell-to-cell variability in Na_V_ current, we performed manual patch clamp measurements on 11 cells. We now include statistics on the mean and standard deviation in maximum Na_V_ current (subsection “Construction and characterization of Na_V_1.7 Optopatch Spiking (Na_V_1.7-OS) HEK cells”, second paragraph).

Finally, the maximal plateau potential evoked by optogenetic stimulation looks dangerously close to spike threshold in Figure 2—figure supplement 1. This aspect must be quantified extensively.

We think the axes in this graph may have been confusing because we had only included a scale-bar, not a complete axis on the inset. We reformatted the display to show clearly that the maximum plateau potential, -26 mV, is well above the spike threshold, -48 mV. We have never encountered a difficulty in optically triggering cells to spike when the cells have adequate Na_V_ capacity.

Additionally, while establishing a proof of principle for all-optical screening using this new optogenetic toolset, the paper falls short of comparing quantitatively all-optical measurements with electrophysiological measurements.

We now include a quantitative calibration of spike height to Na_v_ current density (Figure 2—figure supplement 2). We also note that Figure 5 contain manual patch clamp validation of the hits from our optical screen.

Furthermore, the applicability to a wide range of voltage-gated channels, other than sodium channels, is not clearly demonstrated, given the dependence of the current assay on regenerative spiking.

Indeed, this technique will not work for all ion channels. We demonstrated function for Na_V_1.7, Na_V_1.5 (Figure 5) and K_v_ 4.3 (Figure 6). In the Discussion we now describe possible application to Ca_V_ channels and delayed rectifier potassium channels, hERG and K_V_7 (last paragraph). We believe that Na_V_ channels are a sufficiently important class of targets that assays devoted to them will have an important impact. We changed the Introduction to emphasize that our focus is in Na_v_ channels (last paragraph).

3) The authors point rightly (Discussion, fourth paragraph) that interference of tested pharmacological compounds with the optogenetic reporters and actuators may produce false positive and false negative results, but that it is unlikely to happen. The burden of proof should not be left on potential users of their technique. The interference with the optogenetic tools of a wide subsets of the compounds used in this study, from different chemical families, should be tested using systematic patch-clamp recordings.

We developed an optical assay to determine whether compounds used in the study had off-target effects on Kir2.1, CheRiff, or QuasAr2 (subsection “High throughput screening of Na_V_1.7 inhibitors”, eighth paragraph, new Figure 5—figure supplement 2). The new Figure 5—figure supplement 2 and 2B describes this assay and the result. Seven of the eight tested compounds showed less than 7% change compared to TTX alone, while carbamazepine slightly decreased QuasAr2 signal amplitude.

We also performed systematic patch clamp recordings to validate the optical assay and to characterize the effects of a set of eight test compounds on our optogenetic reporter and actuator (subsection “High throughput screening of Na_V_1.7 inhibitors”, ninth paragraph, Figure 5—figure supplement 2). None of the compounds affected CheRiff photocurrents. Seven of the eight compounds had no effect on QuasAr2 voltage sensitivity, while carbamazepine slightly decreased QuasAr2 voltage sensitivity, consistent with the optical assay for off-target effects. The small off-target effect of carbamazepine had no effect on our optical assays of activity-dependent block (Figure 3 and Figure 3), because the analysis is insensitive to an overall scaling of the signal amplitude.

We believe that the development of an all-optical assay for off-target effects is an important addition to the paper.